# Pre-Clinical Investigations of the Pharmacodynamics of Immunogenic Smart Radiotherapy Biomaterials (iSRB)

**DOI:** 10.3390/pharmaceutics15122778

**Published:** 2023-12-14

**Authors:** Michele Moreau, Shahinur Acter, Lindokuhle M. Ngema, Noella Bih, Gnagna Sy, Lensa S. Keno, Kwok Fan Chow, Erno Sajo, Oscar Nebangwa, Jacques Walker, Philmo Oh, Eric Broyles, Wilfred Ngwa, Sayeda Yasmin-Karim

**Affiliations:** 1Department of Radiation Oncology, Brigham and Women’s Hospital, Dana-Farber Cancer Institute, Harvard Medical School, Boston, MA 02115, USA; bihnoella@gmail.com (N.B.); syasmin-karim@bwh.harvard.edu (S.Y.-K.); 2Department of Radiation Oncology & Molecular Radiation Sciences, Johns’ Hopkins Hospital, Baltimore, MD 21287, USA; sacter1@jh.edu (S.A.); lindokuhle.ngema@students.wits.ac.za (L.M.N.); gsy1@jhu.edu (G.S.); 3Department of Chemistry and Department of Physics (Medical Physics), University of Massachusetts Lowell, Lowell, MA 01854, USA; kwokfan_chow@uml.edu (K.F.C.); erno_sajo@uml.edu (E.S.); 4Department of Pharmacy & Pharmacology, WITS Advanced Drug Delivery Platform Research Unit, University of the Witwatersrand, Johannesburg 2050, South Africa; 5Department of Health Administration and Human Resources, The University of Scranton, Scranton, PA 18510, USA; lkeno1@jh.edu; 6Nanocan Therapeutics Corporation, Princeton, NJ 08540, USA; nebangwa871@gmail.com (O.N.); jwalker@nanocan.life (J.W.); ebroyles@nanocan.life (E.B.)

**Keywords:** anti-CD40 monoclonal antibody, smart radiotherapy biomaterials, toxicity, hematology, histopathology, clinical chemistry

## Abstract

The use of an immunogenic smart radiotherapy biomaterial (iSRB) for the delivery of anti-CD40 is effective in treating different cancers in animal models. This study further characterizes the use of iSRBs to evaluate any associated toxicity in healthy C57BL6 mice. iSRBs were fabricated using a poly-lactic-co-glycolic-acid (PLGA) polymer mixed with titanium dioxide (TiO_2_) nanoparticles incorporated into its matrix. Animal studies included investigations of freely injected anti-CD40, anti-CD40-loaded iSRBs, unloaded iSRBs and control (healthy) animal cohorts. Mice were euthanized at pre-determined time points post-treatment to evaluate the serum chemistry pertaining to kidney and liver toxicity and cell blood count parameters, as well as pathology reports on organs of interest. Results showed comparable liver and kidney function in all cohorts. The results indicate that using iSRBs with or without anti-CD40 does not result in any significant toxicity compared to healthy untreated animals. The findings provide a useful reference for further studies aimed at optimizing the therapeutic efficacy and safety of iSRBs and further clinical translation work.

## 1. Introduction

The steady growth of the immune-oncology field has been evident over the last few decades. Immunotherapy has been used in combination with many different treatment modalities, such as radiation therapy, chemotherapy, or surgery as it pertains to cancer treatment [1]. Immunotherapy’s aim is to engage the immune system to destroy cancer cells, including the utilization of immune-checkpoint-targeted monoclonal antibodies (mAbs) [2]. However, once the immune system has been stimulated, the challenge that often persists is the impermeable microenvironment of the ‘cold’ tumor, which can block the immune T cells’ infiltration [3,4]. Therefore, the immune receptor named agonist mAbs that targets the cluster of differentiation 40 (CD40) presents a possible approach allowing the better facilitation of the infiltration of immune cells within the tumor microenvironment [5]. As a member of the tumor necrosis factor (TNF) receptor family, CD40 is expressed by antigen-presenting cells (APCs), among which are dendritic cells (DCs), which are crucial to provoke antigen-specific T cells in immunoregulation [6]. Agonistic anti-CD40 monoclonal antibodies (mAbs) have shown compelling therapeutic benefits pre-clinically for solid tumors in combination with radiotherapy, chemotherapy, or other immunoadjuvants [3,7]. The anti-tumor effect of agonistic anti-CD40 is both T-cell-driven and T-cell-independent, driven through reprogrammed anti-tumor macrophages [8,9]. However, systemic toxicity is a limiting factor preventing anti-CD40 mAbs from successful clinical translation [10]. New drug delivery approaches that can leverage the anti-tumor activity of anti-CD40 but minimize toxicity are needed. 

The improvement of nanotechnology has generated remarkable growth in the solicitation of nanoparticles and other nanoparticle-based technologies used for drug delivery systems [11]. One such technology is immunogenic smart radiotherapy biomaterials (iSRBs), which incorporate titanium dioxide (TiO_2_) nanoparticles to provide image guidance during radiotherapy. TiO_2_ nanoparticles (size = 1–100 nm) have been extensively used in industrialized and customer goods due to their intensity, extremely high refractive index, and vigorous reactionary activity [12]. Titanium dioxide accounts for 70% of the overall bulk of pigments being manufactured globally, and it has been highly requested in pharmaceuticals, medicines, and food products, among other uses [13,14,15]. TiO_2_ nanoparticles are presently manufactured in abundance due to their great stability and anti-caustic and photocatalytic properties [16]. In nanomedicine, TiO_2_ nanoparticles are examined as valuable implements in nanotherapeutics and cutting-edge imaging [17].

Several studies have argued that titanium dioxide in its natural state has very low toxicity and insignificant biological effects [18,19,20]. The United States Food and Drug Administration (FDA) has accepted TiO_2_ as a food colorant with the requirement that it cannot surpass 1% by body weight (BW) [21]. TiO_2_ was also approved as a substance placed into contact with food, including food packaging and its components, processing equipment, food preparation surfaces, or cookware, by the United States FDA [21]. Nanoparticles have been formulated for medical applications using biodegradable polymers and/or metal oxide particles such as titanium dioxide (TiO_2_) to locally deliver drugs in situ. The combination of drugs with NPs can boost their buildup in infected sites, in addition to their infiltration capabilities through microorganism sheaths [22]. Therefore, several recent studies have been focused on producing drug delivery systems that are polymer-based, such as poly-lactic-co-glycolic acid (PLGA), to deliver drug payloads in situ [23,24]. One such design is the immunogenic smart radiotherapy biomaterials (iSRBs), which present a great opportunity for investigation. 

The effectiveness of iSRBs has been demonstrated in several studies when combined with radiotherapy in different animal models [25,26,27]. One advantage of using iSRBs is that they can directly deliver the anti-CD40 (mAb) into the tumor sub-volume, which is expected to minimize systemic toxicity. Moreover, this approach delivers smaller amounts of anti-CD40 than systemic administration. The iSRBs present a viable pathway for clinical translation since they are radiopaque and could replace currently used fiducial markers for image-guided radiotherapy for some patients [26,27]. Since such inert biomaterials are already being used clinically, the use of iSRBs, which can perform the function of fiducials but also deliver drugs like anti-CD40, presents an excellent opportunity for research and development in cancer treatment. 

A comprehensive toxicity evaluation is necessary before any drug product’s clinical translation, to address any potential side effects of the administered drug. The safety and tolerability of a drug product, such as the iSRB loaded with immunotherapeutic drugs, must be investigated and optimized for clinical translation. Therefore, this work examines the relative toxicity and safety of the iSRB loaded with an anti-CD40 monoclonal antibody in healthy C57BL6 male and female mice. The results are compared to those in control/untreated (healthy) cohorts. 

## 2. Materials and Methods

### 2.1. Fabrication of Immunogenic Smart Radiotherapy Biomaterial (iSRB)

Smart radiotherapy biomaterials were fabricated based on previously published protocols [25,26]. First, 600 mg/mL of poly-lactic-co-glycolic acid (PLGA) (M.W.: 50–50 kDa) and acetone (Sigma-Aldrich, St. Louis, MO, USA) were vortexed to create a homogenous mix. The PLGA hydrogel mix was equally aliquoted in a 1:1 ratio in 2 oz amber jars. Each jar containing the PLGA hydrogel was used to add titanium dioxide (TiO_2_ Anatase, 99.5%, 5 nm) solution (US Research Nanomaterials Inc., Houston, TX, USA) in the ratio of 1:1/9, respectively, to form a nano-hydrogel colloid. The nano-hydrogel colloid was then loaded into silicone tubing (W.W. Grainger Inc., Lake Forest, IL, USA) (inner diameter: 1/16″) using a Harvard apparatus device (Harvard Bioscience, Holliston, MA, USA) at a constant flow rate of 0.001 mL/min. Afterwards, the loaded tubing was cured in a 50 °C oven for up to 96 h. Once cured, the tubing was cut to the desired length, similar in dimensions to the currently used fiducials in the clinic. The iSRB seeds were UV sterilized using a CoolCLAVE Plus personal sterilizer (Cat#: E330220) purchased from AMSBIO (Cambridge, MA, USA).The average mass of iSRBs used in this study fell within 13–14 mg, with an average length of 5 mm and width of 1.5 mm. Some iSRBs were loaded with 20 µg anti-CD40 (Clone FGK4.5, BioXCell, Lebanon, NH, USA). A 10 uL Hamilton digital syringe was used to close the distal end of the iSRB with the nano-hydrogel colloid and it was allowed to cure for 5 min on dry ice. Next, 2.3 µL of the anti-CD40 solution was loaded into the proximal end of each iSRB, followed by the sealing of the proximal end with the nano-hydrogel colloid administered with a 10 uL Hamilton syringe (Sigma-Aldrich, St. Louis, MO, USA). The iSRB could be administered subcutaneously using a brachytherapy needle (TeamBest and Best Medical International, Springfield, VA, USA). Once administered, the iSRB degrades over time for up to 60 days post-implantation according to the molecular weight of the PLGA, the ratio of lactic to glycolic acid in the PLGA, and many other factors. As the iSRB biodegrades, it releases the drug payload, and the polymer components make the tumor micro-environment more immunogenic, making iSRBs suitable for combined radiotherapy and immunotherapy. 

### 2.2. Toxicity Studies in Healthy Mice

#### 2.2.1. Study Design

Eight-week-old wild-type C57BL6 male and female mice were purchased from Taconic Biotech. Ten- to twelve-week-old non-tumor-bearing (healthy) mice were randomized into different groups. Animals in one group were administered the iSRB loaded with anti-CD40 (20 µg). A second group was administered an unloaded iSRB. Other groups investigated for comparison included groups with control healthy animals with no treatment, and, in some cases, the direct injection of anti-CD40 (20 µg) was used. All types of treatment were administered subcutaneously. All mice were maintained, and all studies were conducted following the Dana Farber Cancer Institute (DFCI) IACUC-approved SOPs and protocol (protocol #15-040) in male (*n* = 60) and female (*n* = 48) mice. A similar study was also conducted at Johns Hopkins University in male (*n* = 50) and female (*n* = 50) C57BL6 mice purchased from both Taconic Farms and Charles River simultaneously at 6–8 weeks and aged to 8–10 weeks at the start of treatment. The groups of C57BL6 mice, male and female (*n* = 100), were allocated to different treatment groups: (1) iSRB loaded with anti-CD40, (2) empty iSRB, (3) control with no treatment. Mice were maintained and all studies followed the Johns Hopkins University ACUC-approved standard of practice under protocol #MO21M281. Figure 1 (Adapted from “Protocols and Methodologies” by BioRender.com (2023), Retrieved from https://app.biorender.com/biorender-templates, accessed on 5 December 2023) highlights the study design assessing the pharmacodynamics of iSRB unloaded/loaded anti-CD40 compared to the healthy cohort of mice that received no treatment. 

#### 2.2.2. Body Score and Body Weight Analyses

Healthy mice’s body weights and scores for signs of distress (hunching, lethargy, and ruffled fur) were investigated. The body weights and scores were plotted for each time point observed. 

#### 2.2.3. Liver and Kidney Function and Serum Total Cholesterol Analyses

Blood sampling was done via the vena cava vein, which is perceived as a more sterile blood collection method to assess hepatic and renal function parameters (alanine transaminase (ALT), aspartate aminotransferase (AST), total bilirubin, gamma-glutamyl transferase (GGT), glucose, blood urea nitrogen (BUN), and creatinine) in female (*n* = 3/group) and male (3–4/group) mice at 7 days and 45 days post-treatment, respectively, comparing the no treatment and iSRB-loaded anti-CD40 groups. All collected blood was kept on ice. The serum was separated within one hour after collection by centrifuging the blood at 3500 rpm for 15–20 min in a cold room, following a standard protocol, and saved at −80 F until sent for analysis. Another set of liver and kidney functions were assessed at different time points (days 1, 14, 30, and 90) post-treatment using the cardiocentesis sampling method for both male and female mice that received either no treatment, one implant of empty iSRB, or iSRB loaded with 20 µg of anti-CD40. All collected serum samples were sent to VRL Laboratories (San Antonio, TX, USA) for analysis. 

#### 2.2.4. Complete Blood Count (CBC) Analysis

CBC hematology was performed for a variety of parameters, including white blood cells (WBC), red blood cells (RBC), red blood cell width (RDW%), and hemoglobin (HGB), among other CBC parameters, at the time points of 3 h and days 1, 4, 7, 14, 30, and 90 post-treatment following cardiocentesis blood sampling. In addition, whole blood was collected from mice using vena cava blood sampling for, respectively, seven and forty-five days post-treatment, to evaluate red blood cells (RBC), hemoglobin (HGB), monocyte (MONO) count, hematocrit (HCT%), mean corpuscular hemoglobin (MCH), and mean corpuscular volume (MCV). Whole blood samples were sent for analysis to VRL Laboratories (San Antonio, TX, USA), within 24 h after collection.

#### 2.2.5. Histopathology Analysis

Histopathological analysis of the heart (after collecting blood from cardiocentesis), lungs, kidney, liver, urinary bladder, adrenal gland, mesenteric lymph node, and pancreas of treated and untreated non-tumor-bearing mice was performed at different post-treatment time points: days 1, 7, 14, 30, 56, and 90. Histopathological analyses on the non-punctured heart muscle, via collection from the vena cava vein, were performed in two groups (*n* = 5/group), comparing only healthy mice to treated mice with iSRB-loaded anti-CD40 antibody, at days 7 and 45 post-treatment. Tissue samples collected were fixed in 10% formalin and sent to VRL Laboratories (San Antonio, TX, USA), for histopathology assessments. 

#### 2.2.6. Colitis Assessment 

To identify the possibility of iSRB-anti-CD40-induced colitis, a 3-day post-treated part of the large intestine (including cecum) was resected from euthanized animals and fixed with 10% formalin. To identify the influx of inflammatory cell (neutrophil, monocyte, and granulocyte) infiltration, immune-histological staining (IHC) was performed using the LyG6 antibody in paraffin-embedded colon sections (4 mm). The tissue processing and immunostaining was performed by iHisto (Salem, MA, USA) and quantification of infiltrated LyG6 positive cells was performed with ImageJ Fiji version 2.14.0.

#### 2.2.7. Statistics

Two-way ANOVA followed by Tukey’s multiple comparisons test was performed using GraphPad Prism version 9.5.1 for Windows, GraphPad Software, San Diego, CA, USA, www.graphpad.com accessed during 1st of March to May of 2023. Analyzed data were deemed significant if their *p*-values were within the following ranges: (* *p* < 0.05), (** *p* < 0.01), (*** *p* < 0.001), (**** *p* < 0.0001). 

## 3. Results

### 3.1. Body Weight and Body Score

To measure the mice’s body scores and body weights, they were allocated to their different groups before treatment: (1) no treatment or control (healthy mice); (2) direct subcutaneous injection of anti-CD40; (3) unloaded iSRB implanted subcutaneously; (4) iSRB loaded with anti-CD40 implanted subcutaneously. Each mouse was weighed at the start of the study and subsequently at each pre-determined collection time point. Figure 2a displays the mice’s weights for up to 4 weeks at each time point in male mice for all four groups. A replicate of this study (groups 1, 2, and 4) showed the male and female mice’s weights up to 90 days post-treatment (Figure 2b,c), highlighting that the collection time up to 4-weeks post-treatment did not affect the mice’s weight. In both studies, no significant changes were observed in the mice’s weight over time up to 30-days post-treatment. Appendix A shows no variations in the mice’s body scores at the different collection time points of 3 h, day 1, and weeks 1, 2, and 4 post-treatment. 

### 3.2. Renal and Hepatic Function

Immunocompetent, wild-type C57BL/6 male and female healthy mice were used to conduct toxicity studies. Clinical chemistry analysis assessed many parameters contributing to renal and hepatic function, highlighted in Appendix A (*n* = 3/group), following cardiocentesis blood collection techniques. The vena cava blood sampling method was also used to evaluate a few parameters corresponding to hepatic (Figure 3) and renal (Figure 4) function in female and male C57BL/6 mice at 7 and 45 days post-treatment, respectively, comparing the no treatment and iSRB-loaded anti-CD40 groups (*n* = 3–4/group). 

#### 3.2.1. Hepatic Function

Several of the hepatic function tests are highlighted in Figure 3. Following vena cava vein blood sampling, parameters such as aspartate aminotransferase (AST), gamma-glutamyl transferase (GGT), total bilirubin, and alanine aminotransferase (ALT) were assessed in the samples collected on days 7 and 45 post-treatment for female and male mice, respectively. Figure 3 highlights normal levels for both the non-treated and treated cohorts, showing that the values for all cohorts fell within the reference ranges given for each parameter. 

#### 3.2.2. Renal (Kidney) Function and Serum Cholesterol Levels

The vena cava blood sampling method assessed renal function parameters, glucose, blood urea nitrogen (BUN), and creatinine for the samples collected on days 7 and 45 post-treatment for female and male mice, respectively. Normal reference range levels were observed for all three parameters assessed across both groups (Figure 4). All parameters were within the noted reference ranges across all cohorts.

Different blood sampling techniques, such as cardiocentesis or vena cava blood, were applied at different time points throughout this study to assess the liver and kidney function. The hepatic and renal function results following the cardiocentesis blood sampling method are shown for the different collection time points of days 1, 14, 30, and 90 in Appendix A for all the male and female mice, respectively. Appendix A display the mean data values within the given reference values for each parameter tested, corresponding to every cohort of mice.

### 3.3. Hematological Analyses

Immunocompetent, wild-type C57BL/6 male and female healthy mice were used to conduct hematology studies. Cell blood count analysis was conducted to screen and evaluate autoimmune toxicity, hematologic condition, or pathology associated with the iSRB treatment. Following blood collection via cardiocentesis or the vena cava vein, complete cell blood count parameters were evaluated, and the generated values from cardiocentesis can be seen in Appendix A. The complete blood cell count parameters, highlighted in Appendix A, corresponding to 3 h and days 1, 4, 7, 14, 30, and 90 post-treatment, indicate comparable values between the treated and non-treated groups. No abnormalities were observed that would indicate immune activation/inflammation. Following the vena cava blood sampling, a few hematological parameters were highlighted, as shown in Figure 5 for female and male mice at days 7 and 45 post-treatment. Comparable values across all cohorts were found, within the reference ranges given for each parameter.

### 3.4. Histopathology Report

Histopathological analysis of the heart, lungs, both kidneys, spleen, liver, urinary bladder, adrenal gland, mesenteric lymph node, and pancreas of treated and untreated non-tumor-bearing mice was performed at different post-treatment time points: days 1, 4, 7, 14, 30, 56, and 90. Initially, small cohorts of male and female mice (*n* = 1–6) were used to generate a pathology report, assessing any differences between treatment groups compared to the control (healthy) group at 10 weeks post-treatment (Appendix A). Appendix A describes the histological findings in selected organs from non-tumor-bearing mice at different time points post-treatment. Figure 6a–d display the percentage of mice tissue over the different time points, shown in Appendix A, with no lesions distinguishing female and male mice. Lesions can be described as any damage or injury incurred by an organ/tissue (e.g., perivascular lymphocytic aggregates, micro-abscesses, adrenal cortex, zona fasciculata lipidosis). Figure 6 and Appendix A represent the scoring and degree of lesions incurred depending on the treatment group. Figure 6c displays significant differences (* *p* < 0.05) between male and female mice across all cohorts in the percentage of mice tissue that showed no lesions. Among the tissue collected, the liver showed significantly (*** *p* < 0.001) more lesions compared to the heart, lung, spleen, and both kidneys (Figure 6d). However, there was no statistically significant difference in lesions incurred by the liver between cohorts. An ulcerative colitis assessment (Figure 6e) was performed to determine any bowel inflammation across all cohorts. No significant difference in LyG6+ immune cell infiltration was observed in the empty iSRB and iSRB–anti-CD40 cohort compared to the control (no treatment) group. 

Appendix A display the pathology reports corresponding to days 7 and 45 post-treatment for female (*n* = 5) and male (*n* = 5) mice used for the vena cava blood sampling. Most lesions were found in the liver tissue compared to the kidneys in both the no treatment (healthy) and treated cohorts of mice. 

## 4. Discussion

Early clinical trial phase 1 studies [29,30] have reported that a single inoculation of the agonistic anti-CD40 monoclonal antibody (mAb) in patients with advanced solid tumors can achieve immune stimulation. Given the potential for hepatotoxicity associated with the intravenous or subcutaneous injection of the anti-CD40 monoclonal antibody, the success of this approach may require sustained local delivery. The results in this study provide a helpful reference when using iSRBs loaded anti-CD40 mAbs. These initial results will inform ongoing work to optimize such an approach for clinical translation. 

The application of immunogenic smart radiotherapy biomaterials represents a practical approach to providing image guidance and sustained drug release during treatment [26,27]. This study assessed toxicity at pre-determined time points after one single injection of either the anti-CD40, empty iSRB, or iSRB loaded with anti-CD40 monoclonal antibody. Blood urea nitrogen (BUN) levels from mice treated with iSRB–anti-CD40 were found to be within the normal range and comparable with the levels from the untreated mice on days seven and forty-five post-intervention of the study (Figure 4 and Appendix A). This suggests that the kidneys were unaffected by either the iSRB–anti-CD40 or empty iSRB. Likewise, the creatinine levels remained within the normal range (Figure 4 and Appendix A), without any significant deviations in the recorded levels between the treatment groups and the control. 

The gamma-glutamyl transferase (GGT), total bilirubin, AST, and ALT levels for both non-treated mice and those injected with iSRB–anti-CD40 remained within the normal range on days seven and forty-five post-treatment, as shown in Figure 3. Additionally, normal levels of blood cell parameters were observed, as shown in Figure 5 and Appendix A, across all cohorts. Laboratory values within normal ranges were seen in male mice treated with iSRB unloaded/loaded with anti-CD40 compared to similar cohorts of female mice and control (no treatment) mice (Appendix A). Gender-based differences in the phenotypes of resident leukocytes [31,32] have been observed in C57BL6 mice.

In addition, the vena cava blood sampling method was performed for two cohorts of mice, comparing no treatment to the iSRB-loaded anti-CD40 group in assessing hepatic and renal function parameters. All tested parameters were within the reference range (Figure 3 and Figure 4). The pathology report shown in Appendix A further corroborates the minimal lesions found in the kidneys and livers of the non-treated and treated cohorts. Figure 6a–d quantify the number of lesions incurred by all the groups of male and female mice, showing a significant difference between the sexes. The difference in the number of lesions incurred by male versus female mice may be due to many factors, such as physiological metabolism, which exhibits a distinction among the sexes because of the inherently diverse biology of the two sexes [31,33]. Consequently, this study only investigated one administration of the iSRB-loaded anti-CD40 implant subcutaneously in mice, and did not investigate the impact of adding one fraction of low-dose radiation or intravenous injection of anti-CD40 to compare their associated toxicity. However, numerous studies have investigated the associated toxicity generated from systemic treatment with an agonistic anti-CD40 preclinically or clinically [34,35]

Standard variations in rodents’ clinical chemistry [36] and hematology [37] parameters transpire due to multiple effects, including the strain, age, gender, nutrition, restriction, anesthesia, day-to-day effects, lodging method, dormant infections, blood collection hemorrhage site, and handling methods [38]. These mentioned features contribute significantly to the obtained outcomes. This study’s limitations consist of the method of mice euthanization, which can impact the observed serum chemistry and hematology values [39,40,41]. However, animals were euthanized similarly, and no significant differences were observed. Additional areas of investigation include varying the anti-CD40 dose and the number of administered iSRBs, and the optimization of the iSRB formulation parameters. Successful optimization of the iSRB technology for clinical translation could provide an innovative platform combining image-guided radiotherapy and immunomodulators, such as anti-CD40, to improve treatment outcomes.

## 5. Conclusions

This pre-clinical investigation suggests that implanting iSRBs with and without anti-CD40 results in no significant toxicity compared to healthy untreated animals. This establishes a solid reference for the further development of this approach, combining image-guided radiotherapy with local immunotherapy delivery. The advantage of this approach also includes administering smaller amounts of the monoclonal antibody in comparison to systemic treatment, which may further decrease the risk of toxicity. The use such iSRBs instead of the currently used inert radiotherapy biomaterials like fiducial markers could be applied at no additional inconvenience for many cancer patients. It could provide a novel modality to augment treatment responses.

## Figures and Tables

**Figure 1 pharmaceutics-15-02778-f001:**
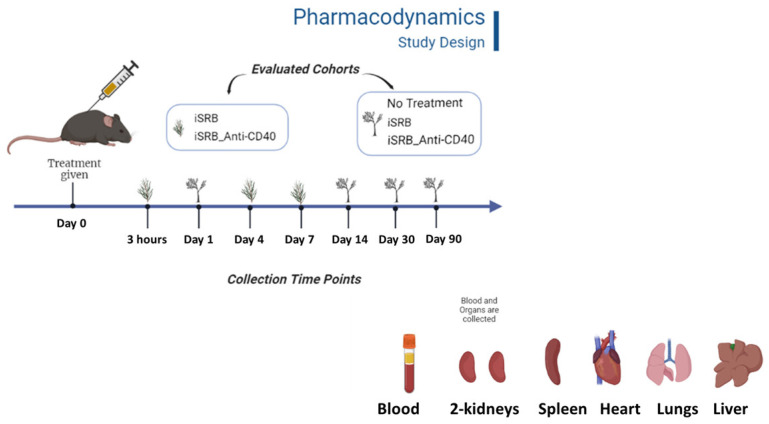
Pharmacodynamics study design in healthy C57BL6 male and female mice. The groups of C57BL6 mice, male and female (*n* = 100), were allocated to different treatment groups: (1) iSRB loaded with anti-CD40 (*n* = 37), (2) empty iSRB (*n* = 26), (3) direct injection of anti-CD40 (*n* = 6), and (4) control with no treatment (*n* = 37). Tissues collected for histopathology report included lung, two kidneys, spleen, heart, and liver. Cardiocentesis blood collection was mostly used to collect blood from mice. Whole blood and serum were collected for the clinical chemistry and hematology analyses. Mice’s body weight was measured at each collection time point mentioned in the illustration. Study design for the initial pharmacodynamics study conducted in male and female mice where additional organs (urinary bladder, pancreas, mesenteric lymph node, and adrenal gland) were collected for the pathology report.

**Figure 2 pharmaceutics-15-02778-f002:**
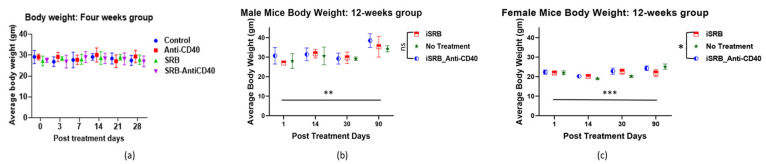
Mice (male and female) body weights are displayed in the plots above for each time point of observation. (**a**) The mice’s body weights (*n* = 3/group) up to 4 weeks post-treatment (male or female mice) show no weight variation. Similar trends are seen in (**b**) male and (**c**) female mice body weights (*n* = 3/time points) for up to 90 days. However, mice body weights displayed in Figure 2b (** *p* < 0.01) and 2c (*** *p* < 0.001) were significantly different between day 1 and day 90 post-treatment. In addition, there was a significant (* *p* < 0.05) difference in the female mice body weight across all cohorts tested. There were minimal to no variations were observed in mice weights across all cohorts for each time points evaluated in the plots shown. ns = not significant.

**Figure 3 pharmaceutics-15-02778-f003:**
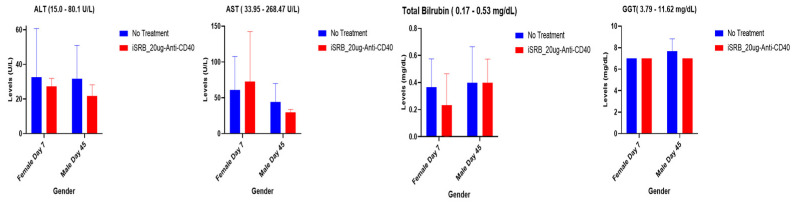
Hepatic function parameters following the vena cava blood sampling method corresponding to day 7 and 45 post-treatment, respectively, with the annotated reference range for several parameters noted in the title of each plot. Only two groups were assessed: (1) no treatment (*n* = 3–4) and (2) iSRB-loaded anti-CD40_20ug (*n* = 3–4) in male and female mice. There was no difference between the no treatment group and the iSRB-loaded anti-CD40 group for both male and female mice.

**Figure 4 pharmaceutics-15-02778-f004:**
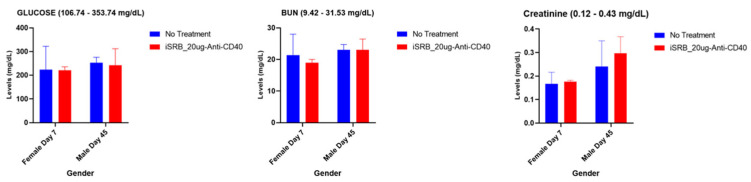
Renal function parameters following the vena cava blood sampling method were assessed for a few parameters in female and male mice, respectively, at 7 and 45 days post-treatment. No treatment (*n* = 3–4) group was compared to the iSRB-loaded anti-CD40-20µg (*n* = 3–4) group. All values were relatively within the noted reference range for all groups in male and female mice.

**Figure 5 pharmaceutics-15-02778-f005:**
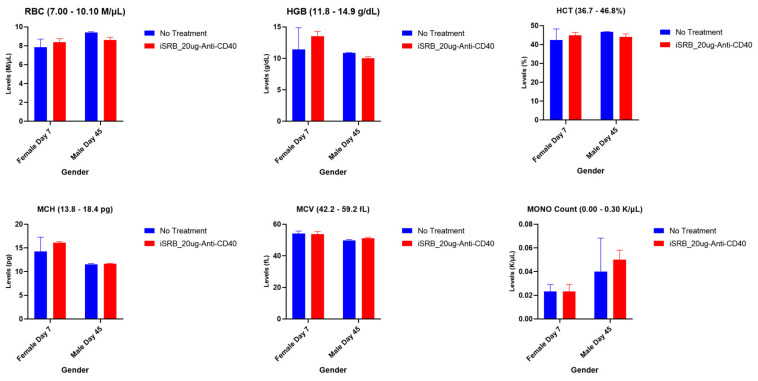
Cell blood count parameters corresponding to days 7 and 45 post-treatment for female (*n* = 3) and male (*n* = 4) mice, respectively, with the annotated reference range for each parameter noted in the title of each plot. Comparable RBC, HGB, HCT, MCH, MCV, and Mono count levels were observed between the no treatment (healthy) group and the one that received the iSRB-loaded anti-CD40 implant subcutaneously, and the values were within the noted reference range for each parameter.

**Figure 6 pharmaceutics-15-02778-f006:**
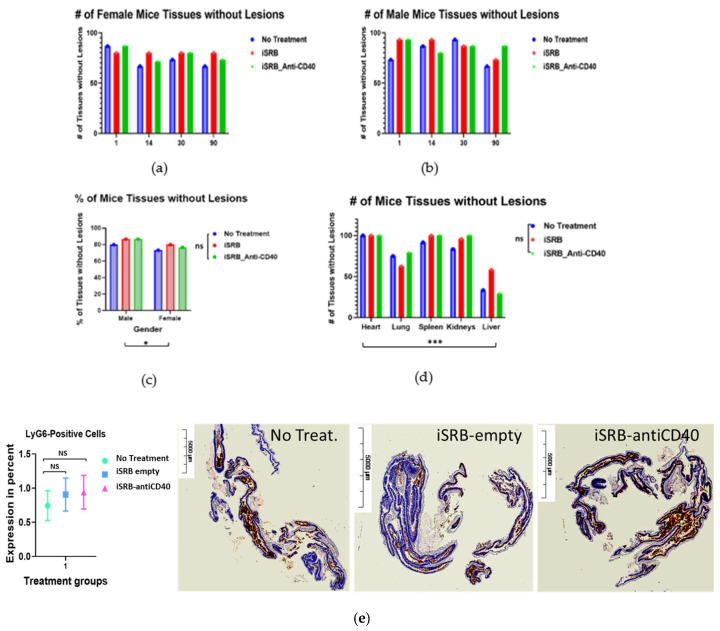
Histopathology examination was performed to evaluate any incurred injuries in different organs collected from healthy untreated mice, treated with the unloaded iSRB or iSRB loaded with anti-CD40 antibody at the time points of days 1, 14, 30, and 90 post-treatment. (**a**,**b**) More mice without lesions were found across all cohorts. (**c**) Both male (*n* = 3/group/time point) and female (*n* = 3/group/time point) mice showed a significant (* *p* = 0.0198) amount of tissue (>73%) with no damage. (**d**) The liver in all the cohorts showed significantly (*** *p* = 0.0003) greaternumber of lesions compared to the other tissues. Lesions can be any damages incurred by the organ, such as Hemorrhage intra-alveolar (likely iatrogenic, due to the method of euthanasia) or Hepatocellular swelling, with intracytoplasmic microvesicular lipidosis. Most mice tissues incurred no lesions except the liver across all cohorts. The common lesions found in the liver include (1) Microgranulomas, characterized by small nodular accumulations (up to 30 cells) of lymphocytes, histiocytes, a few neutrophils, and entrapped degenerating hepatocytes; or (2) Microabscesses, characterized by small nodular accumulations of neutrophils, lymphocytes, and histiocytes (up to 30 cells) with entrapped degenerating hepatocytes. (**e**) No iSRB–anti-CD40-induced colitis was observed three days post-treatment, confirming the infiltrated neutrophil count. Immunotherapy induced colon inflammation was assessed by histological technique. IHC staining for LyG6+ inflammatory cell was performed in colon mucosa treated with empty iSRB, iSRB–anti-CD40, and PBS treated controls (*n* = 3). Quantification of the severity of colon wall inflammation was calculated by LyG6+ inflammatory cells infiltration in treated mice colon mucosa on day 3 post-treatment. Ly6G is a good marker for detection of early inflammatory granulocytes like neutrophils and monocytes [28]. NS/ns = not significant.

## Data Availability

Data are contained within the article or Appendix A. Data are also available on request from the corresponding author.

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
