# Peer review of "Pre-Clinical Investigations of the Pharmacodynamics of Immunogenic Smart Radiotherapy Biomaterials (iSRB)"

_pharmaceutics, 2023, doi:10.3390/pharmaceutics15122778_

Round 1

Reviewer 1 Report

Comments and Suggestions for Authors

Presented manuscript devoted to preclinical evaluation of biodegradable material for prolonged release of anti CD40 antibodies, potentiating antitumor immune response. I have some minor revisions prior the manuscript can be accepted:

1) Why all the parameters were shown only for 3 mice per group? In Supplementary materials, Tabsle S5 information is presented for 5 mive per group. Why these animals were not included in main text? Moreover on Figure 1 it is written that for iSRB loaded mice n=30, empty iSRB n =24, pure anti CD-40 n=6, healthy control n=30. Where are all these data? Why only 3 mice per group were shown? How they were chosen?

2) What is release kinetics of anti-CD40 antibodies for biodegradable matrix?

3) The name of the article is positioning radiotherapy application for these materials. If so, please provide all data for irradiated animals for all groups, otherwise i suggest to remove information about radiotherapy from name of the manuscript

4) Why  for ALT/AST/billirubine there is so big deviation even for untreated animals? What is the range of normal values for these enzymes for healthy animals? Maybe the "normal" range can be included as dashed lines on diagrams?

5)Please describe how the loading of antibodies was performed.

6) Please indicate the location of adminstration of materials and thier dosage (mass of all construction, not only amount of antibodies)

7) In Materials and Methods section it is writeen that pure anti-CD40 anti bodies were used as a control. For groups with pure anti-CD40 antibodies and  with pure iSRB no data except  Body weight are presented. Why these data are not shown if the design is showing 4 groups? 

Author Response

Presented manuscript devoted to preclinical evaluation of biodegradable material for prolonged release of anti CD40 antibodies, potentiating antitumor immune response. I have some minor revisions prior the manuscript can be accepted:

  1. Why all the parameters were shown only for 3 mice per group? In Supplementary materials, Table S5 information is presented for 5 mive per group. Why these animals were not included in main text? Moreover on Figure 1 it is written that for iSRB loaded mice n=30, empty iSRB n =24, pure anti CD-40 n=6, healthy control n=30. Where are all these data? Why only 3 mice per group were shown? How they were chosen?

The total number of mice used in the entire study has been reflected in the main text mentioned in method section 2.2.1 Study Design. However, these are studies that have been conducted in 3 years’ time, therefore, the main text is showing data from vena cava method for male and female mice. The supplementary materials contain the data for all of the animals used for each particular studies of hepatic and renal functions, hematology and pathology reports.

2) What is release kinetics of anti-CD40 antibodies for biodegradable matrix?

This manuscript is only focused on how safe the iSRB unloaded or loaded anti-CD40 are in healthy cohorts of male and female mice. The release kinetics study of this anti-CD40 antibody from the biodegradable matrix has not been done for our iSRB seed but will be conducted in the near future.

3) The name of the article is positioning radiotherapy application for these materials. If so, please provide all data for irradiated animals for all groups, otherwise i suggest to remove information about radiotherapy from name of the manuscript

Several manuscripts mentioned in the main text have been published on this iSRB detailing its use as an image guided fiducial marker that can be used during radiotherapy to locate the tumor for external beam radiation. Henceforth, that is why it’s called immunogenic smart radiotherapy biomaterials (iSRBs). We do acknowledge that none of the treatment arms included an irradiated group but that would be investigated in future studies. Resource: Mueller, R.; Moreau, M.; Yasmin-Karim, S.; Protti, A.; Tillement, O.; Berbeco, R.; Hesser, J.; Ngwa, W. Imaging and Characterization of Sustained Gadolinium Nanoparticle Release from Next Generation Radiotherapy Biomaterial. Nanomaterials 202010, 2249. https://doi.org/10.3390/nano10112249

4) Why for ALT/AST/billirubine there is so big deviation even for untreated animals? What is the range of normal values for these enzymes for healthy animals? Maybe the "normal" range can be included as dashed lines on diagrams?

The normal range values have been included. Despite apparent deviations, the values fall within normal range.

5) Please describe how the loading of antibodies was performed.

The loading of the antibodies have been added to the manuscript reflected in lines 80 – 85.

6) Please indicate the location of adminstration of materials and thier dosage (mass of all construction, not only amount of antibodies)

From lines 84 – 85: The iSRB could be administered subcutaneously using a brachytherapy needle (TeamBest, VA, USA). Line 79: Some iSRBs were loaded with 20µg anti-CD40.  Lines 84 - 85: The iSRB could be administered subcutaneously using a brachytherapy needle. From Lines 77 – 78: The average mass of iSRBs used in this study fall within 13-14mg with an average length of 5mm and width of 1.5mm.

7) In Materials and Methods section it is writeen that pure anti-CD40 antibodies were used as a control. For groups with pure anti-CD40 antibodies and  with pure iSRB no data except  Body weight are presented. Why these data are not shown if the design is showing 4 groups? 

The Study Design section 2.2.1 described the different cohorts of mice used throughout the different studies. Animals in one group were administered iSRB loaded with anti-CD40 (20 µg). A second group was administered with unloaded iSRB. Other groups investigated for comparison included groups with direct injection of anti-CD40 (20 µg) and a control healthy animal group with no treatment.

However, the freely injected anti-CD40 group was only used in the initial studies evaluating the body weights and scores and the histological findings ten-weeks post-treatment. The raw data are shown for all of these cohorts in the supplementary materials.

Reviewer 2 Report

Comments and Suggestions for Authors

The article conducted an assessment of the toxicity associated with immunogenic smart radiotherapy biomaterial (iSRB) and their counterparts loaded with therapeutic anti-CD40 (iSRB_anti-CD40). Healthy C57BL6 mice received subcutaneous injections of synthesized iSRBs containing PLGA-TiO2 nanoparticles for a comprehensive toxicity evaluation, which included analyses of blood chemistry, cell blood counts, and pathological imaging. The results of the study indicate that whether iSRBs contain anti-CD40 or not, they do not induce significant toxicity when compared to untreated healthy animals. However, it is important for the paper to present more detailed data and provide a thorough discussion to substantiate this conclusion.

1.     Please provide the characterization of iSRB and anti-CD40-loaded iSRB, including information such as size, zeta potential, anti-CD40 loading efficiency, and other relevant parameters for a comprehensive understanding.

2.     Please clarify the rationale behind selecting 20 μg of anti-CD40 as the dosing amount to justify the chosen dosage.

3.     It would be beneficial if the authors could discuss the potential use of iSRB_Anti-CD40 for intravenous (IV) injection and compare it to subcutaneous (SC) injection in terms of potential toxicity differences.

4.     On page 393, kindly include a citation for the paper that provides the reference range of the tested parameters to support the claims made in the study.

5.     The study primarily focuses on single-dose toxicity evaluation. It would be valuable to address the potential implications of multiple injections and their impact on toxicity.

Author Response

The article conducted an assessment of the toxicity associated with immunogenic smart radiotherapy biomaterial (iSRB) and their counterparts loaded with therapeutic anti-CD40 (iSRB_anti-CD40). Healthy C57BL6 mice received subcutaneous injections of synthesized iSRBs containing PLGA-TiO2 nanoparticles for a comprehensive toxicity evaluation, which included analyses of blood chemistry, cell blood counts, and pathological imaging. The results of the study indicate that whether iSRBs contain anti-CD40 or not, they do not induce significant toxicity when compared to untreated healthy animals. However, it is important for the paper to present more detailed data and provide a thorough discussion to substantiate this conclusion.

  1. Please provide the characterization of iSRB and anti-CD40-loaded iSRB, including information such as size, zeta potential, anti-CD40 loading efficiency, and other relevant parameters for a comprehensive understanding.

From Lines 77 – 78: The average mass of iSRBs used in this study fall within (13 – 14) mg with an average length of 5mm and width of 1.5mm. Anti-CD40 loading process has been added. Zeta potential was not characterized in this study for the antibody was obtained commercially as described in the methods and materials section. Further details of the iSRB have been published in previous work (Mueller, R.; Moreau, M.; Yasmin-Karim, S.; Protti, A.; Tillement, O.; Berbeco, R.; Hesser, J.; Ngwa, W. Imaging and Characterization of Sustained Gadolinium Nanoparticle Release from Next Generation Radiotherapy Biomaterial. Nanomaterials 2020, 10, 2249. https://doi.org/10.3390/nano10112249)

  1. Please clarify the rationale behind selecting 20 μg of anti-CD40 as the dosing amount to justify the chosen dosage.

20ug of anti-CD40 was the amount that have been used in the previous studies (Yasmin-Karim, S.; Ziberi, B.; Wirtz, J.; Bih, N.; Moreau, M.; Guthier, R.; Ainsworth, V.; Hesser, J.; Makrigiorgos, G.M.; Chuong, M.D.; et al. Boosting the Abscopal Effect Using Immunogenic Biomaterials With Varying Radiation Therapy Field Sizes. Int. J. Radiat. Oncol. 2022, 112, 475–486, doi:10.1016/j.ijrobp.2021.09.010) where the efficacy of the drug product of iSRB loaded anti-CD40 was evaluated and thus the same amount of anti-CD40 was assessed for toxicity.

  1. It would be beneficial if the authors could discuss the potential use of iSRB_Anti-CD40 for intravenous (IV) injection and compare it to subcutaneous (SC) injection in terms of potential toxicity differences.

The iSRB_Anti-CD40 is a seed as described in previous works cited within the manuscript such as reference#: 6, 7, and 8. Therefore, it would not be suitable for intravenous (IV) injection. In addition, this study evaluates the associated toxicity from implanting a drug product such as the iSRB-loaded anti-CD40, which would biodegrade sustainably and release the antibody over time. There would be no use for a subsequent injection.

  1. On page 393, kindly include a citation for the paper that provides the reference range of the tested parameters to support the claims made in the study.

VRL Maryland Inc. provides the reference range as the core facility that analyzed the given samples to generate these results. Jason Lankasky, the laboratory manager at VRL-MD, stated: “The reference values given for each parameter are based upon data from our instrument pertaining to non-study mice run in house. This is however not specific to any particular strain, sex, or age category.” VRL Maryland Inc. can be contacted to provide citations. As for now, here is somce citation: Otto GP, Rathkolb B, Oestereicher MA, Lengger CJ, Moerth C, Micklich K, Fuchs H, Gailus-Durner V, Wolf E, HrabÄ› de Angelis M. Clinical Chemistry Reference Intervals for C57BL/6J, C57BL/6N, and C3HeB/FeJ Mice (Mus musculus). J Am Assoc Lab Anim Sci. 2016;55(4):375-86. PMID: 27423143; PMCID: PMC4943607.

  1. The study primarily focuses on single-dose toxicity evaluation. It would be valuable to address the potential implications of multiple injections and their impact on toxicity.

The idea is to do one injection of this iSRB for sustained delivery of the anti-CD40 drug over time and not multiple doses, as this is the route for clinical translation. This study can be used as a reference for larger animal studies that will be conducted in dogs or monkeys in the near future where different numbers of iSRBs would be administered.

Reviewer 3 Report

Comments and Suggestions for Authors

1. Line 80: ‘’the iSRB degrades over time…” How long it takes for this degradation?

2.  Why was day 7 post-treatment selected to be the reference for the study?

3. It should be mentioned the total number of mice (male and female) used for the entire study at the beginning in the Study Design not only in Fig. 1 for coherence of the reading. Why only to 6 it was a direct injection of Anti-CD40, how many male and female?

4. As written in the article line 211-214, hepatic and renal functions parameters were evaluated in female mice? Why not in male too to see possible differences?

5. Regarding the histopathology report and Fig. 6c (line 307-309) what’ s authors explanation for the significant difference?

Comments on the Quality of English Language

-

Author Response

Comments and Suggestions for Authors

  1. Line 80: ‘’the iSRB degrades over time…” How long it takes for this degradation?

From Line 85 – 87: Once administered, the iSRB degrades over time for up to 60-days post implant according to molecular weight of PLGA and ratio of Lactic to glycolic acid of the PLGA and many other factors based on our experiment. As the iSRB biodegrades, it releasing releases the drug payload. Previous studies have highlighted the biodegradation over time of iSRB protoypes via imaging (Mueller, R.; Moreau, M.; Yasmin-Karim, S.; Protti, A.; Tillement, O.; Berbeco, R.; Hesser, J.; Ngwa, W. Imaging and Characterization of Sustained Gadolinium Nanoparticle Release from Next Generation Radiotherapy Biomaterial. Nanomaterials 2020, 10, 2249. https://doi.org/10.3390/nano10112249).

  1. Why was day 7 post-treatment selected to be the reference for the study?

Most of the time points mentioned in the supplementary data were done using cardiocentesis (3hours, days 1, 4, 7, 14, 30 and 90) as the method of blood collection and some of the values obtained for the hepatic and renal functions’ parameters were skewed even for the control group of healthy mice. Therefore, days 7 and 45 were chosen as a quick way to check if using the vena cava blood sampling method would help to get normal values that fall within the range that VRL Maryland Inc Facility annotated as the reference range for each hepatic and renal function parameters.

  1. It should be mentioned the total number of mice (male and female) used for the entire study at the beginning in the Study Design not only in Fig. 1 for coherence of the reading. Why only to 6 it was a direct injection of Anti-CD40, how many male and female?

We have made the necessary changes in section 2.2.1 Study design. The n = 6/group refers to supplementary materials Table S4 only. This study is using a new approach of loading the anti-CD40 in iSRB which will sustainably release the drug over time and therefore, the remainder of the mice mentioned in the study design section only investigated the no treatment, empty iSRB, or iSRB loaded anti-CD40 cohorts.

  1. As written in the article line 211-214, hepatic and renal functions parameters were evaluated in female mice? Why not in male too to see possible differences?

The data for male mice was gathered at a later time point day 45 which we have shown in Figures 3, 4, 5 and supplementary materials Tables S6 - S7.

  1. Regarding the histopathology report and Fig. 6c (line 307-309) what’ s authors explanation for the significant difference?

The authors’ explanation have been added in Lines 448 – 457.

Round 2

Reviewer 1 Report

Comments and Suggestions for Authors

Authors have answered all my comments

Reviewer 2 Report

Comments and Suggestions for Authors

The authors have responded to all the queries raised by the reviewer, making it suitable for acceptance in its present form.

Reviewer 3 Report

Comments and Suggestions for Authors

accepted in the revised form